# Enabling dynamic modelling of coastal flooding by defining storm tide hydrographs

Job C. M. Dullaart[1], Sanne Muis[1,2], Hans de Moel[1], Philip J. Ward[1], Dirk Eilander[1,2], Jeroen C. J. H. Aerts[1,2]

1. Institute for Environmental Studies (IVM), Vrije Universiteit Amsterdam, Amsterdam, The Netherlands
2. Deltares, Delft, The Netherlands

*Correspondence to:* Job C. M. Dullaart (job.dullaart@vu.nl)

## Abstract

Coastal flooding is driven by the combination of (high) tide and storm surge, the latter being caused by strong winds and low pressure in tropical and extratropical cyclones. The combination of storm surge and the astronomical tide is defined as the storm tide. To gain understanding into the threat imposed by coastal flooding and to identify areas that are especially at risk, now and in the future, it is crucial to accurately model coastal inundation. Most models used to simulate coastal inundation scale follow a simple planar approach, referred to as bathtub models. The main limitations of this type of models are that they implicitly assume an infinite flood duration and they do not capture relevant physical processes. In this study we develop a method to generate hydrographs called HGRAPHER, and provide a global dataset of storm tide hydrographs based on time-series of storm surges and tides derived with the global tide and surge model (GTSM) forced with the ERA5 reanalysis wind and pressure fields. These hydrographs represent the typical shape of an extreme storm tide at a certain location along the global coastline. We test the sensitivity of the HGRAPHER method with respect to two main assumptions that determine the shape of the hydrograph, namely the surge event sampling threshold and coincidence in time of the surge and tide maxima. The hydrograph dataset can be used to move away from planar to dynamic inundation modelling techniques across different spatial scales.

## 1 Introduction

Over the course of the 21[st] century, coastal populations increasingly at risk of flooding due to sea level rise (SLR) (Oppenheimer et al., 2019). In addition, the number of people living in coastal areas below 10 m elevation worldwide is projected to increase from over 600 million people today to more than 1 billion people by 2050 under all Shared Socioeconomic Pathways scenarios (Merkens et al., 2016), which means that the exposure will increase. Global coastal flood risk assessments can help identifying areas that are potentially exposed to flooding under both current and future climate conditions (Ward et al., 2015). To setup these flood risk assessments, it is important to understand the dynamics of storm surges generated from strong winds and low pressure in tropical (TCs) and extratropical cyclones (ETCs) and how these generate coastal flooding (Resio and Westerink, 2008). Flood models can be used to model these coastal inundation dynamics resulting from extreme storm tides, where the storm tide is defined as the combination of storm surge and the tide (Colle et al., 2010).

Coastal inundation models have varying levels of complexity. Global models all follow a simple planar approach (Brown et al., 2018; Dullaart et al., 2021a; Kirezci et al., 2020; Lincke and Hinkel, 2018; Muis et al., 2016). These models, often referred to as bathtub models, assume that any land that is below

a specific static water level and that is connected to the sea will be inundated. The main limitation of
the planar approach is that it assumes an infinite flood duration (e.g. temporal evolution of a storm
surge) and does not capture the physical hydrodynamic processes that drive coastal flooding. This can
be partly addressed by accounting for water-level attenuation (Vafeidis et al., 2019; Haer et al., 2018;
Tiggeloven et al., 2020). Local to regional-scale models generally apply a (hydro)dynamic modelling
approach that captures the physical processes that drive flooding (Lewis et al., 2013; Pasquier et al.,
2019; Vousdoukas et al., 2018). Model comparisons at regional scale have shown that in terms of flood
extent and depth the dynamic modelling approach is more accurate than the planar approach
(Ramirez et al., 2016; Vousdoukas et al., 2016a). Generally, the planar approach overestimates the
flood extent due to the assumption that flood propagation is only limited by topography, and that high
water levels are maintained for an infinite duration (Stephens et al., 2021). The main reasons for
applying the planar approach across different spatial scales, instead of the dynamic approach, are the
simplicity of setting up a planar model, low computational costs, and limited requirements for input
data.
Due to the advances in high-performance computing and the development of reduced-physics
dynamic inundation models (Leijnse et al., 2021; Yin et al., 2016; Bates et al., 2010), there is the
potential to improve flood mapping across different spatial scales and step away from using the planar
approaches for coastal inundation modelling. First applications of dynamic inundation models at
continental scale have been published (e.g. Vousdoukas et al., 2016a). However, flood maps are often
derived for a specific return period (RP), for example a flood map corresponding to the 1 in 100-year
water level. While planar models only need information about the height of the extreme water level,
dynamic models also need information about the duration. The temporal evolution of an extreme
water level, composed of tide and surge, is referred to as the hydrograph (Chbab, 2015; Sebastian et
al., 2014; Salisbury and Hagen, 2007). Throughout this study we use the term hydrograph to refer to
the storm tide hydrograph. Hydrograph characteristics that determine the flood severity are, among
others, the maximum storm tide level, base duration, and overall shape. For example, when the water
level is elevated for a longer period of time, particularly close to the time of high water when defence
exceedance is most likely, the water will propagate further inland (Santamaria-Aguilar et al., 2017;
Quinn et al., 2014). Currently, a global dataset of hydrographs that can be applied for dynamic
inundation modelling for specific RPs is lacking. Vousdoukas et al. (2016) made a first step towards
dynamic inundation modelling at the continental-scale for Europe. In this study, the temporal
evolution of extreme water levels is incorporated by the use of a generic empirical formulation. The
surge hydrograph is assumed to be an isosceles triangle with a duration based on a linear fit
relationship between modelled surge heights and the half event duration. In reality the rising and
falling limb of the surge hydrograph can have a distinct shape that have different durations, and varies
from location to location (MacPherson et al., 2019). The tidal component in Vousdoukas et al., (2016)
is represented by taking the highest tidal level from a 10-year simulation. Instead, a time-varying value
could be used to include tidal variation, including the spring-neap cycle, in a more accurate way. While
some advances have been made in modelling storm tide hydrographs, the current understanding of
the temporal evolution of sea levels during extremes is limited.
The aim of this study is to address this research gap by developing and applying a globally-applicable
method (HGRAPHER) to generate hydrographs. In doing so, we pave the way for coastal flood mapping
using dynamic models across different spatial scales. First, we review the various methods available
to define a hydrograph and their main assumptions. Second, building on existing literature, we present
the open-source HGRAPHER method with a global dataset of hydrographs for 23,226 locations along
the world's coastline. As input, we use 38 years of storm surge and tide simulations (1979-2018)
derived with the Global Tide and Surge Model (GTSM) forced with the ERA5 climate reanalysis (Muis
et al., 2020). Third, the sensitivity of the HGRAPHER method is tested with respect to two main
assumptions that determine the shape of the hydrograph, namely: 1) using normal high tide or spring
tide; and 2) the coincidence of the surge and tide maximum or a time offset between the two
maximums. Last, we discuss the limitations of our methodology and ways forward.

## 2 Available methods to generate hydrographs

In this section we give an overview of four hydrograph generating methods. The reason for including these studies on hydrographs in this review, from the wide variety of studies that exists on this topic (e.g. Sebastian et al., 2014; Chbab, 2015; Environment Agency, 2018; MacPherson et al., 2019; Vousdoukas et al., 2016a; Xu and Huang, 2014; Salisbury and Hagen, 2007), is that they all have a clearly distinct methodology. Based on this review, we can select the hydrograph generating method that best fits our study goals. All four methods use multi-year water level time series from tide gauge stations or model simulations as input, but they differ in terms of input parameter used, the way the surge hydrograph is computed, and how tide and surge levels are combined. Table 1 summarizes the main characteristics of the four methods.

**Table 1:** Main characteristics of four hydrograph methods

| study | study area | hydrograph method | | |
|---|---|---|---|---|
| | | input parameter | surge hydrograph | combine tide and surge |
| Chbab, 2015 | Dutch coast | surge residual | averaging | linearly |
| Environment Agency, 2018 | United Kingdom coast | skew surge | fit distribution | joint probability method |
| MacPherson et al., 2019 | German Baltic Sea coast | storm tide | parametric | not required |
| Vousdoukas et al., 2016 | European coast | surge and wave setup | best linear fit relationship | constant value for tide |

The first method by Chbab (2015) starts by computing the residual water level. The surge residual is the difference between the predicted tide and the storm tide level (*Fig. 1*). Predicted tides are estimated by harmonic analyses to determine the amplitude and phase of the different tidal constituents. To define the surge hydrograph, events are selected from the residual time series by means of the Peaks-Over-Threshold (POT) method using 1.5 m as a threshold. A 48-hour time window lasting from 24 hours before until 24 hours after the surge maximum is extracted. The final step to obtain the surge hydrograph is normalizing and averaging all 48-hour time series of surge levels. To test the sensitivity of the surge hydrograph to the chosen parameters, a sensitivity analysis is performed. They conclude that the upper 50% of the normalized surge height (normalized surge height > 0.5) is not affected when either the threshold or time window length is increased or decreased. This is an important finding because it indicates that the surge hydrograph is most robust close to the time of high water when defence exceedance is most likely (Santamaria-Aguilar et al., 2017; Quinn et al., 2014). However, a longer time window (of e.g. 72 or 96 hours) results in a longer base duration. The argument given for using a 48-hour time window is that 48 hours is the typical duration of a storm along the Dutch coastline. The surge hydrograph is added linearly to the average tidal cycle where the surge maximum is assumed to coincide with the tide maximum. To generate a hydrograph corresponding to a specific RP, the unitless surge hydrograph is scaled to a certain water level. For example, if the average maximum tide is 1 metre and the 100-year storm tide is 3 metres, the surge hydrograph is multiplied by 2. In areas with a large tidal range and a wide and shallow continental shelf, tide-surge interaction may induce a time offset between the two maxima (*Fig. 1*). For example, in the North Sea the surge maximum generally occurs 2.5 hours before the tidal maximum (Chbab, 2015; Horsburgh and Wilson, 2007). This is because a storm surge increases the depth and thereby modulates the influence of bottom friction and the speed of the tidal wave (Pugh, 1996; Rego and Li, 2010). The time offset can be taken into account by computing the time offset between the surge and tidal maxima for all surge events above the POT 99[th] percentile (POT99). Subsequently, the average offset is used to shift the surge time series relative to the tidal maximum.

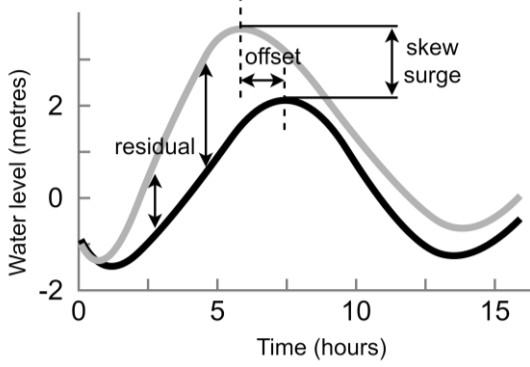


*Figure 1: schematic of the residual, offset, and skew
surge. Time series of the tide (grey line), and the tide
including meteorological effects (black line) are shown.*

The second method, developed by the U.K. Environment Agency (2018) starts by computing the skew
surge. Skew surge (*Fig. 1*) refers to the difference between the maximum storm tide level and
maximum tidal level within a tidal cycle, irrespective of their timing (Williams et al., 2016). An
important reason for using skew surge instead of the surge residual is that the latter can arise due to
tide-surge interaction (Idier et al., 2019). In contrast to the surge residual, for the skew surge there is
no need to account for timing offsets, apart from some locations where a dependency between skew
surge and high tidal levels is observed (Santamaria-Aguilar and Vafeidis, 2018). To generate the skew
surge hydrograph, the 15 most extreme skew surges are selected. An argument for selecting this
number of events is not given. Both the high and low water skew surge values are extracted for each
storm event. Subsequently, the high and low water skew surge values are interpolated to a 15-minute
timeseries and normalized. Then, the duration of each of the 15 surges at particular percentiles (i.e.
10%, 20% and so on) are calculated. The maximum duration at each percentile is used to compute the
skew surge hydrograph. The study by U.K. Environment Agency does not combine the skew surge
hydrograph with tidal level time series.
The third method by MacPherson et al. (2019), that further developed the method from Wahl et al.
(2011, 2012), starts by identifying storm tide events. To do this, a POT method is used. Using POT is
preferred over annual maxima because the number of events extracted is typically higher with POT
resulting in a more robust representation of the local storm tide characteristics in the hydrograph.
Then, each event is characterized through a parameterization scheme. A total of 17 parameters are
calculated such as peak water level, event duration, and the flow (rising limb) and ebb (falling limb)
curve shape. Subsequently, synthetic hydrographs are generated through Monte Carlo simulations
using the obtained parameters. This means that for a single return period multiple storm tide
hydrographs are available with different shapes but the same maximum water level.
The fourth method by Vousdoukas et al. (2016) starts by computing the high tide water level (HTWL).
The HTWL is calculated as a constant water level that consists of the mean sea level (MSL) and the
maximum tide elevation taken from a 10-year time series. The assumption that the maximum high
tidal level occurs along the entire duration of the event, thereby neglecting tidal variations, can
significantly overestimate the water level in places with large tidal variability, such as north-western
Australia. The HTWL is then combined with time-varying storm surge levels and wave setup to obtain
total water levels. Time series of storm surge levels (1979-2014) are taken from Vousdoukas et al.
(2016b) and wave setup is approximated by 20% of the significant wave height, both based on the
ERA-Interim global climate reanalysis (Dee et al., 2011). To obtain information about the temporal
evolution of an extreme event, extreme events are identified in the available time series of surge and
wave setup. For each identified event the duration and peak water level are extracted. Subsequently,
a best linear fit relationship between the duration and peak water level is estimated. To conclude, the
combined hydrograph consists of the HTWL combined with a symmetric triangle shaped time series
on top of it representing the surge and wave setup for a certain return period.
Comparing the four methods, we find that the hydrograph generating methods that are developed for
application at smaller scales are tailored towards the local water level characteristics. This makes them
less suitable for application at larger scales. For example, in the study by Chbab (2015) a threshold of
0.5 m is used to identify extreme surge events in time series. However, at the global scale surge levels
exceeding 0.5 m do not occur in some regions such as the south of the Caribbean. The hydrograph
generating method developed by U.K. Environment Agency (2018) is developed for regions that
experience a substantial tidal range such as the U.K., as it is based on skew surge values. However, the
complete global coastline does not experience such high tides. In addition, MacPherson et al., (2019)
developed a method that is applicable in areas with a small tidal range, making it well suited for the
German Baltic Sea coast and larger scales such as the entire Baltic Sea, but inapplicable at continental
to global scales. The last study that we discussed (Vousdoukas et al., 2016a) takes a more simple
approach to define hydrographs for continental Europe. The tidal component is represented by a
constant value and is combined with a triangle shaped time-varying storm surge. Overall, the study by
Vousdoukas et al. (2016a) is a step towards modelling inundation at larger scales using hydrographs.
However, substantial improvements can be made to the hydrograph generating method. To this end,
we will build on Chbab (2015) because, most importantly, the method used in this study does take a
time-varying surge and tide component into account. In addition, instead of representing the surge by
a triangle shape in the combined hydrograph like Vousdoukas et al. (2016a), the method from Chbab
(2015) allows the rising and falling limb of the hydrograph to have different shapes. This results in a
more accurate representation of the shape of the storm surge in the combined hydrograph. It is
especially important that the hydrograph represents the water level correctly close to high water
when defence exceedance is most likely, and because the water will propagate further inland if the
water level is elevated for a longer period of time (Santamaria-Aguilar et al., 2017; Quinn et al., 2014).

# 3 Methods


Figure 1 summarizes the main steps of the HGRAPHER hydrograph generating model. Storm tide
levels, tidal time series, and storm tide RPs are used as input. First, extreme events are identified in
the surge time series and used to compute a normalized surge hydrograph. Second, the average tide
signal is computed from the tidal time series, Third, the hydrograph is generated by combining the
average tide signal with the normalized surge hydrograph. To create the final hydrograph, this generic
shape is scaled to an absolute water level height for specific RPs based on the COAST-RP dataset
(Dullaart et al., 2021b).
3.1 Input data

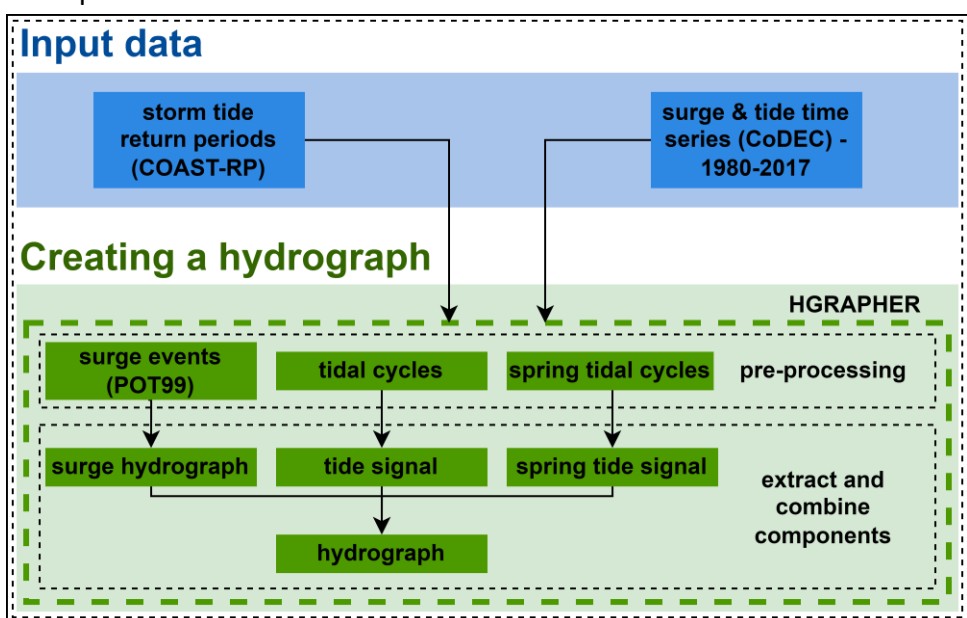

*Figure 2: Modelling framework*

Time series of storm tides (1980-2017) at a 10-minute interval from 23,226 output locations are taken
from the CoDEC-ERA5 dataset (Muis et al., 2020). The CoDEC-ERA5 dataset was generated by forcing
the 2D depth-averaged hydrodynamic Global Tide and Surge Model (GTSM) with wind and pressure
fields from the ERA5 climate reanalysis (Hersbach et al., 2019). GTSM forced with ERA5 has shown to
accurately simulate maximum surge heights of historical TC and ETC events (Dullaart et al., 2020). In
addition, a comparison between modelled and observed annual maxima, showed a mean bias of -0.04
m (with a standard deviation of 0.32 m) (Muis et al., 2020). Overall, the time series of surge and tidal
levels from the CoDEC-ERA5 dataset are of good quality and therefore valid input data to HGRAPHER.
The surge time series are computed as the difference between a storm tide simulation and a tide-only
simulation. As a result, the surge time series include non-linear tide-surge interaction effects
(Horsburgh and Wilson, 2007). The output locations are located at every 50 km along the coastline. In
addition, the locations of tide gauge stations are included. In order to scale the hydrograph to a storm
tide level that corresponds with a certain RP, we use storm tide RPs from the global COAST-RP dataset
(Dullaart et al., 2021b). In contrast to other global storm tide RP datasets, COAST-RP explicitly takes
into account low-probability high impact TCs (Dullaart et al., 2021a) by making use of 3,000 years of
synthetic TC tracks from the STORM dataset (Bloemendaal et al., 2019).

## 3.2 Creating a hydrograph

### 3.2.1 Surge hydrograph

The following procedure is used for each of the 23,226 output locations individually. To generate a hydrograph of the surge, we start with extracting independent extremes from the surge time series based on the Peaks-Over-Threshold (POT) method. Using the POT method for selecting extremes is preferred over annual maxima as the latter could result in excluding extreme events that happened in the same year. We use the 99[th] percentile over the complete time series as threshold and we select peaks that are at least 72 hours apart to ensure independent events (Wahl et al., 2017; Vousdoukas et al., 2016b; Haigh et al., 2016). The threshold results in the selection of on average 1 surge event per year and 40 events over the full time series. Setting the threshold is a trade-off between having an event set of sufficient size to compute a representative average shape without including too many relatively small surge events that would too strongly affect the resulting shape (see section 4.4). For each selected surge event, we first extract the time series from 36 hours before, until 36 hours after the peak (*Fig. 3a*). Second, each 72-hour surge event is normalized (i.e. dividing each surge level by the peak) such that the maximum surge value is equal to 1 (unitless). Third, we combine the selected surge events to calculate the average surge hydrograph. This is done by determining the time (relative to the peak) at which a specific surge height (from 0 to 1 with increments of 0.01) is exceeded. As an example, in Fig. 3a we show that for one surge event the exceedance time at a normalized surge height of 0.25 is 14.0 hours before and 26.0 hours (16.0 + 10.0) after the surge maximum occurred as indicated by the black arrows. Then, for each normalized surge height the average exceedance time is computed, similar to Chbab (2015), resulting in an average curve. Because the shape of the rising and falling limb of the surge can differ, the exceedance time is calculated separately for each, and they are subsequently merged into the final average surge hydrograph.

### 3.2.2 Average and spring tide signal

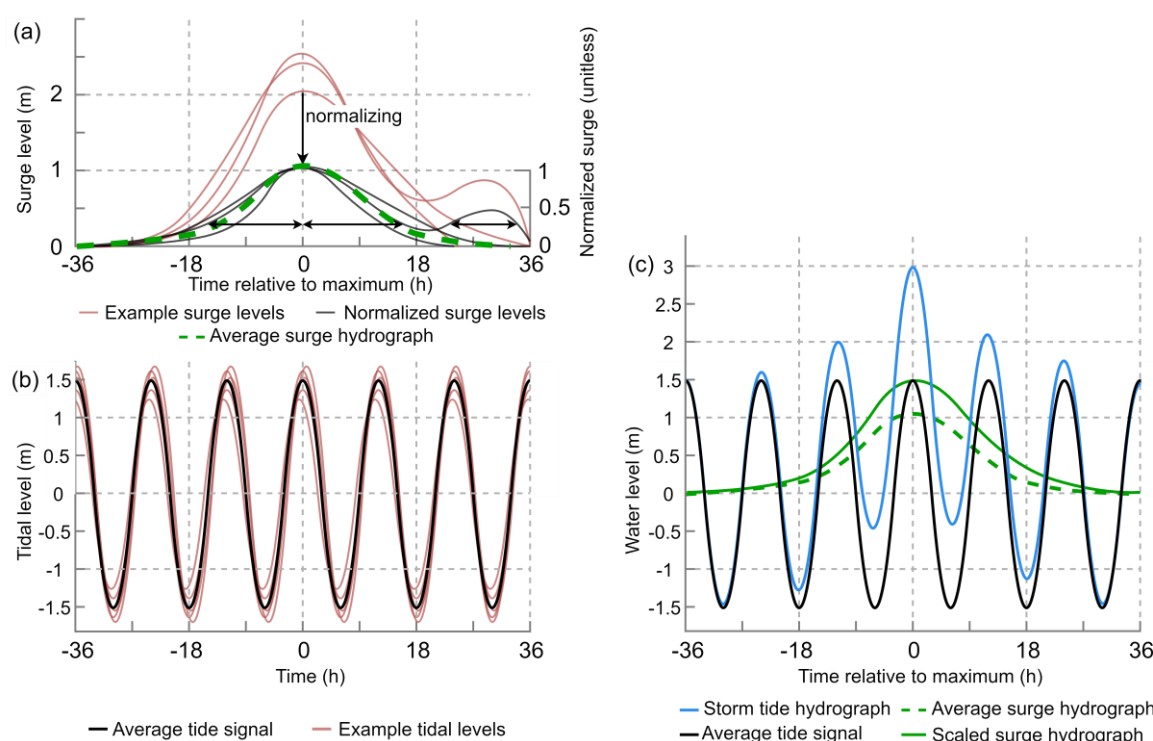

*Figure 3: Visualization of the steps leading to the storm tide hydrograph of a hypothetical 1-in-100 year event of 3.0 m with the a) surge hydrograph, where the black arrows indicates the period over which a normalized surge height of 0.25 is exceeded. Note that for the falling limb we take the sum of the two time periods for which this is the case; b) average tide signal, and c) storm tide hydrograph. The average surge hydrograph is scaled to 1.5 m such that the combined water level equals the 1-in-100 year storm tide level of 3.0 m.*

Next, we combine the surge hydrograph with the average tide signal (*Fig. 3b*). To create a curve
representing the average tide signal we take three steps. First, we split the tidal series from the period
1980-2017 up into segments that are each 24 hours and 50 minutes long. The start and end times of
the tidal cycles are selected from the tide time series by searching for a minimum around 24h and 50
minutes after the previous low tide. The segment length is based on the phase of the M2 tidal
component which is equal to a lunar day (24 hours and 50 minutes). At most locations around the
world the M2 is the main tidal component. Second, we compute the mean over all tidal segments to
obtain the average tide segment. Third, we duplicate the average tide segment to obtain a longer tidal
time series to which we refer as the average tide signal.
In addition, we extract the spring tide signal because a storm surge event happening at spring tide can
result in a very different shape of the hydrograph. The spring-neap tide cycle takes two weeks. To
extract the average spring tide signal, we first search for the highest tide every two weeks. Second,
we select 72 hours of the tidal time series before and after the spring tide maximum. This procedure
is repeated for the available time series of the tide (1980-2017), after which we compute the mean
over all spring tides to extract the average spring tide signal.
3.2.3 Storm tide hydrograph
The surge hydrograph is combined with the average tide or spring tide to create a storm tide
hydrograph (*Fig. 3c*). In theory the surge maximum can coincide with any tide. However, in shallow
regions the timing will be influenced by interaction effects between the surge and the tide which
results in a phase difference. This is for example the case in the North Sea where the tidal wave will
start travelling faster under storm conditions due to the increased water level which reduces the
bottom friction (Horsburgh and Wilson, 2007; Resio and Westerink, 2008). To determine whether a
typical time offset between the surge and tide should be taken into account, we extract the
distribution of the timing offset between the surge and tidal maximum during the most extreme surge
events (POT99). For most locations around the globe, the distribution of the timing offset does not
show a clear signal (see section 4.4). Therefore, we assume that the surge and tidal maximum coincide.
With HGRAPHER a hydrograph can be generated for a total water level of interest. In this study we
use storm tide levels corresponding to a 100-year return period (RP100) because this is an often-used
coastal protection standard (Lamb et al., 2018; FEMA, 1968). If, for example, the RP100 storm tide
level is 3.0 meters and the average high tide is 1.5 meter this means the unitless average surge
hydrograph has to be scaled up to 1.5 meters, such that maximum surge plus the maximum tide is
equal to 3.0 meters (*Fig. 3c*).

# 4 Results

## 4.1 Storm surge hydrographs

For each output location from the CoDEC-ERA5 dataset, a surge hydrograph is generated. For illustration, results are shown for La Rochelle in France and Marco Island in the United States (*Fig. 4a & 4b*). We find a storm surge duration (i.e. the time over which the normalized surge height is above zero) of 54 hours in La Rochelle and 42 hours in Marco Island. Other studies find comparable storm surge durations of 40 hours for Hoek van Holland and 45 hours for Den Helder in The Netherlands (Chbab, 2015), and between 40 and 70 hours for the German Baltic Sea coast (MacPherson et al., 2019). The difference in storm surge duration between La Rochelle and Marco Island is likely caused by the different type of storms occurring in these regions. TCs can cause a fast shift from onshore to offshore winds when making landfall, which results in the surge becoming negative in just a couple of hours. Hurricane Irma is an example of a TC that made landfall near Marco Island and caused such a fast shift in surge levels. The normalized surge level time series have a strong irregular behaviour. This originates from the fact that the surge time series are obtained by subtracting tide-only simulations from a total water level simulations (including tidal and meteorological forcing). Therefore, the surge time series are the residual water level that include tide-surge interaction effects, and we believe this

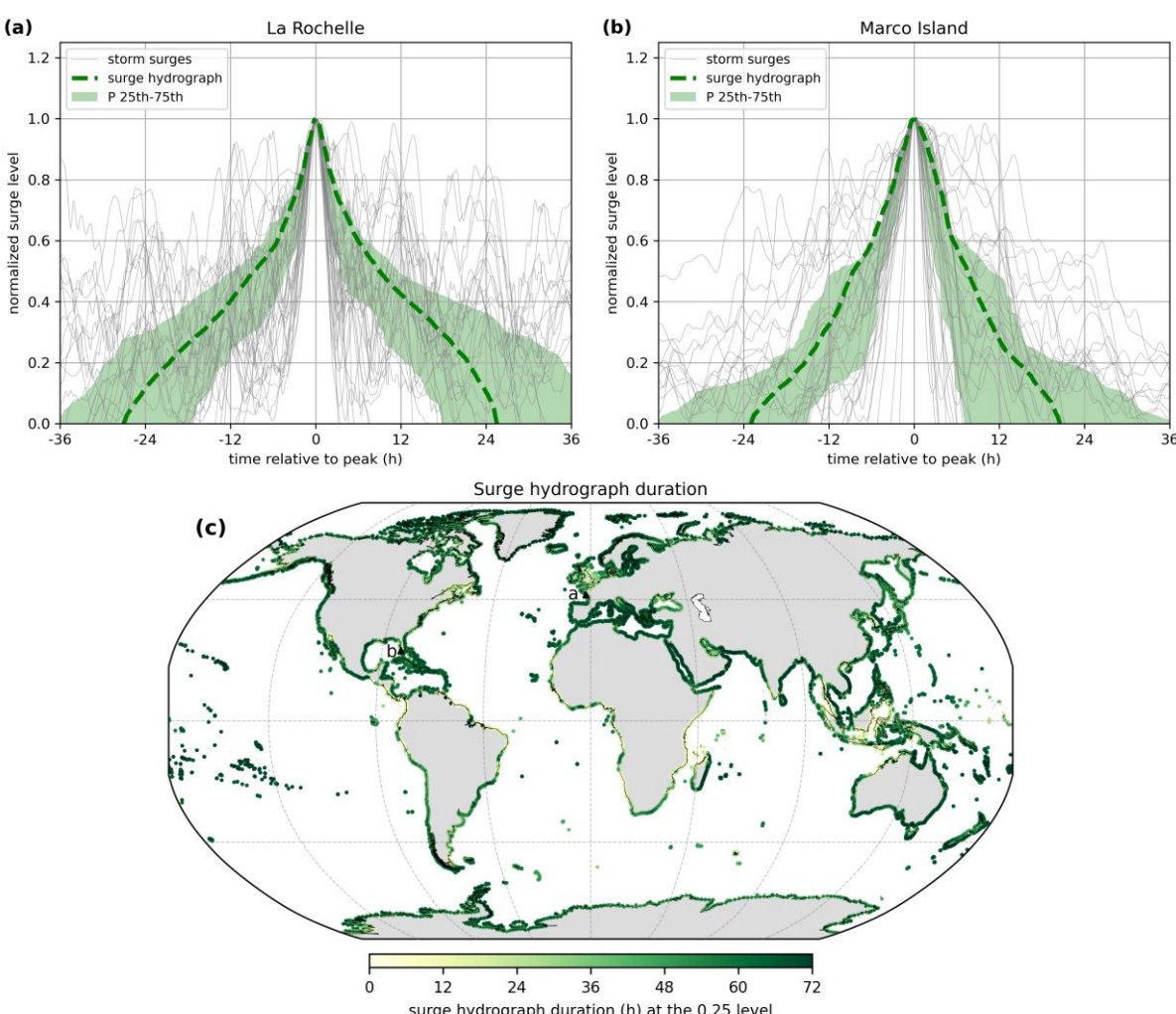

*Figure 4: Surge hydrograph (dashed green line) for a) La Rochelle and b) Marco Island. Normalized surge levels are shown in grey and the green shaded area represents the $25^{th} - 75^{th}$ percentile. Panel c) shows the surge hydrograph duration at 0.25, with the locations of La Rochelle and Marco Island indicated by a and b, respectively.*

partly explains the irregular behaviour. Differences in the evolution of storms over time can also
contribute to the variability observed at the different time steps of the normalized surge levels,
particularly in areas that are affected by TCs and ETCs as the characteristics of the two types of storms
differ considerably (Domingues et al., 2019). In addition, not all 40 events are extreme over their
complete lifetime, which means that noise is affecting the lower ends of the hydrograph. Taking the
mean over the normalized surge heights removes this irregular shape. At the global scale a distinct
pattern shows up in certain regions (*Fig. 4c*). In Europe for example, the average storm surge duration
is substantially lower in the North Sea compared to the Atlantic coastline and the Baltic Sea. Last, we
computed the difference in surge hydrograph duration between the 25[th] and 75[th] percentile at a
normalized surge height of 0.75 (*App. Fig. A1*). This can provide some insights in the variability of flood
duration, assuming that inundation might starts to occur around the 0.75 normalized surge height.
## 4.2 Average (spring) tide signal
For each output location the average and spring tide signal are computed. Although the tidal range at
La Rochelle is substantially larger than at Marco Island, the general shape of the average tide signal is
comparable (*Fig. 5a & 5b*). Both locations show a large variation in amplitude between tidal cycles.
For spring tide, the variation in the tidal amplitude between the tidal cycles is smaller. Note that the

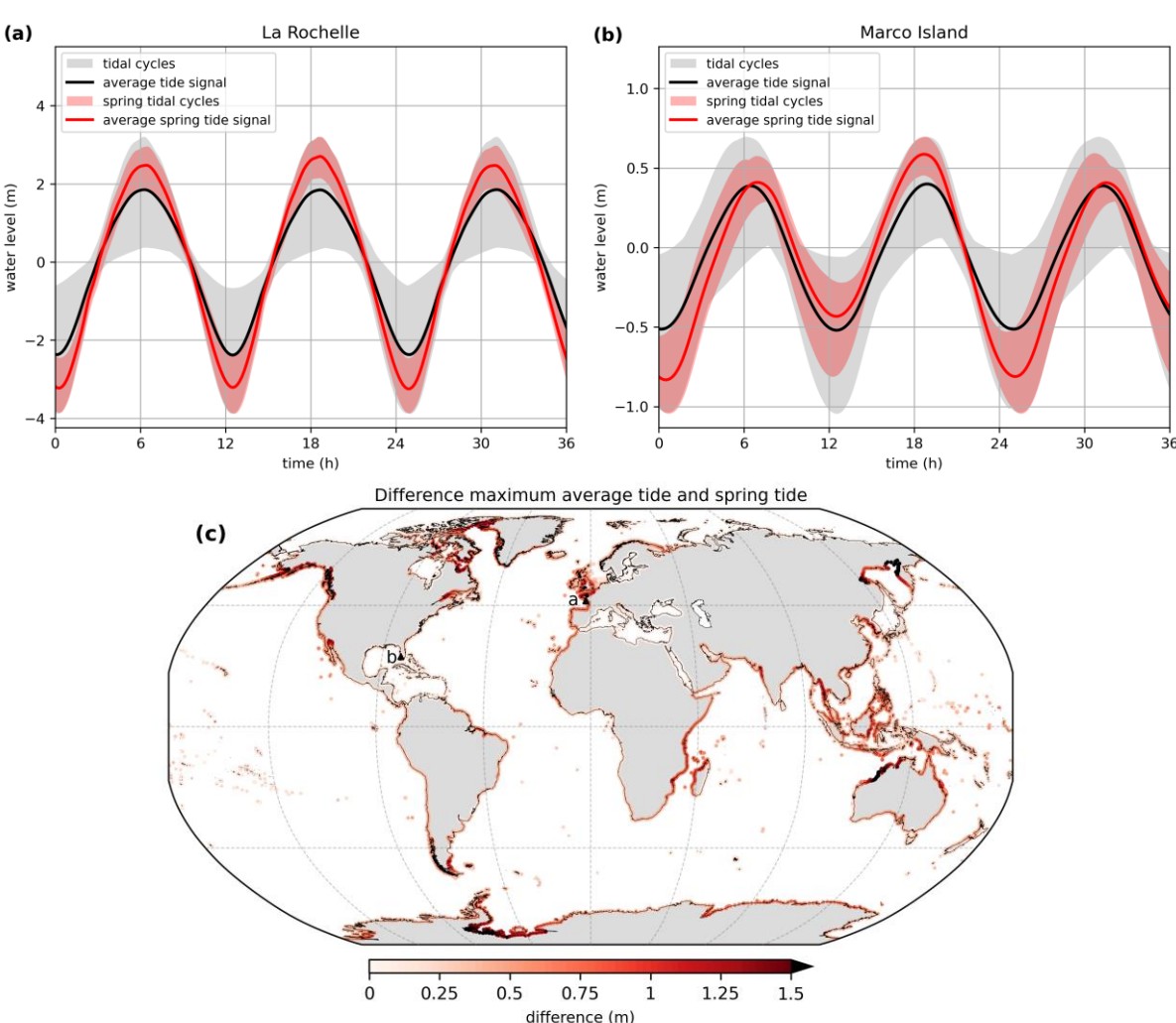

*Figure 5: Average tide signal (black line) for a) La Rochelle and b) Marco Island. The grey shaded area shows the range of all tidal cycles. The average spring tide signal is shown in red and the red shaded area indicate all tidal cycles that are used to compute the average spring tide signal. Panel c) shows the absolute difference between the maximum average tide signal and average spring tide signal. The location of La Rochelle and Marco Island are indicated by the letters a and b, respectively.*

grey shaded area exceeds the red shaded area at both locations during the first and third high tide
because the average spring tide signal is computed by taking the average over a two week period,
while the average tide signal is computed by taking the average over the daily tidal cycle of 24 hours
and 50 minutes. Furthermore, the duration of the first and second high and low tide cycle of a tidal
day differs at Marco Island. This is caused by the type of tide at this location which is a mixed
semidiurnal tide (i.e. a tidal regime with two high and low tides per tidal day of different size) (Song
et al., 2011). Computing the average tide signal can be difficult at locations with a very small tidal
amplitude and mixed semidiurnal tide such a Montevideo (*Fig. A1*). Because of the large number of
shapes that the tidal cycles can have here, taking the average will not completely represent all possible
shapes. However, because the average high tide values are correctly represented by the average
(spring) tide signal, the findings are not affected to a large extent. For La Rochelle the maximum
average tide signal increases 46% from 1.85 m based on all tidal cycles to 2.70 m when taking the
average of the spring tidal cycles. In Marco Island the maximum average tide signal is 0.40 m and the
maximum average spring tide is 0.59 m (+48%). The larger absolute difference in La Rochelle means
that for an extreme storm tide to occur the timing of the surge maximum relative to the (spring) tide
maximum is more important compared to Marco Island. When applying HGRAPHER, it is important to
understand the typical characteristics of a storm tide in the area of interest, because this information
is needed to choose between the average and spring tide signal. For example, in northwest Australia
the difference between the maximum average and spring tide signal exceeds 1.5 m (*Fig. 5c*), indicating
that in this region an extreme storm tide is much more likely to occur during spring tide. Therefore,
using the average spring tide signal should be considered. At the global scale, the difference between
the average and spring tide signal maxima exceeds 0.5 m and 1.0 m, at 24% and 3% of all output
locations, respectively.

## 4.3 Storm tide hydrographs

The surge hydrograph is scaled up to a certain water level and combined with the average tide signal to obtain the storm tide hydrograph (*Fig. 6a and 6b*) that corresponds to the 1-in-100 year (RP100) storm tide level from the COAST-RP dataset (Dullaart et al., 2021b). In La Rochelle the RP100 storm tide level is 3.76 m and the average high tide is 1.85 m. Therefore, the unitless surge hydrograph is scaled up to 1.91 m, such that the combined water level equals the RP100 storm tide level. At Marco Island the RP100 storm tide level is 2.18 m to which the tide contributes 0.40 meter and the surge 1.78 meter, respectively. From the RP100 storm tide hydrograph that we create globally it is possible to deduce the relative contribution of the surge (*Fig. 6c*). Especially in areas where the maximum spring tide signal substantially exceeds (>0.5 m) the maximum average tide signal the surge contribution might be too large compared to observed historical events. This effect is counteracted by the assumption that the surge and tide coincide in time. As a result, a smaller surge is sufficient to get to the desired RP100 storm tide level compared to the situation where a time offset is implemented to combine the average tide signal with the scaled surge hydrograph. Last, the surge hydrographs are based on the surge residual including tide-surge interaction effects. These interaction effects tend to be positive at low tide and negative at high tide (Horsburgh and Wilson, 2007). As a result, we might overestimate the contribution of the surge to the combined hydrograph at high tide.

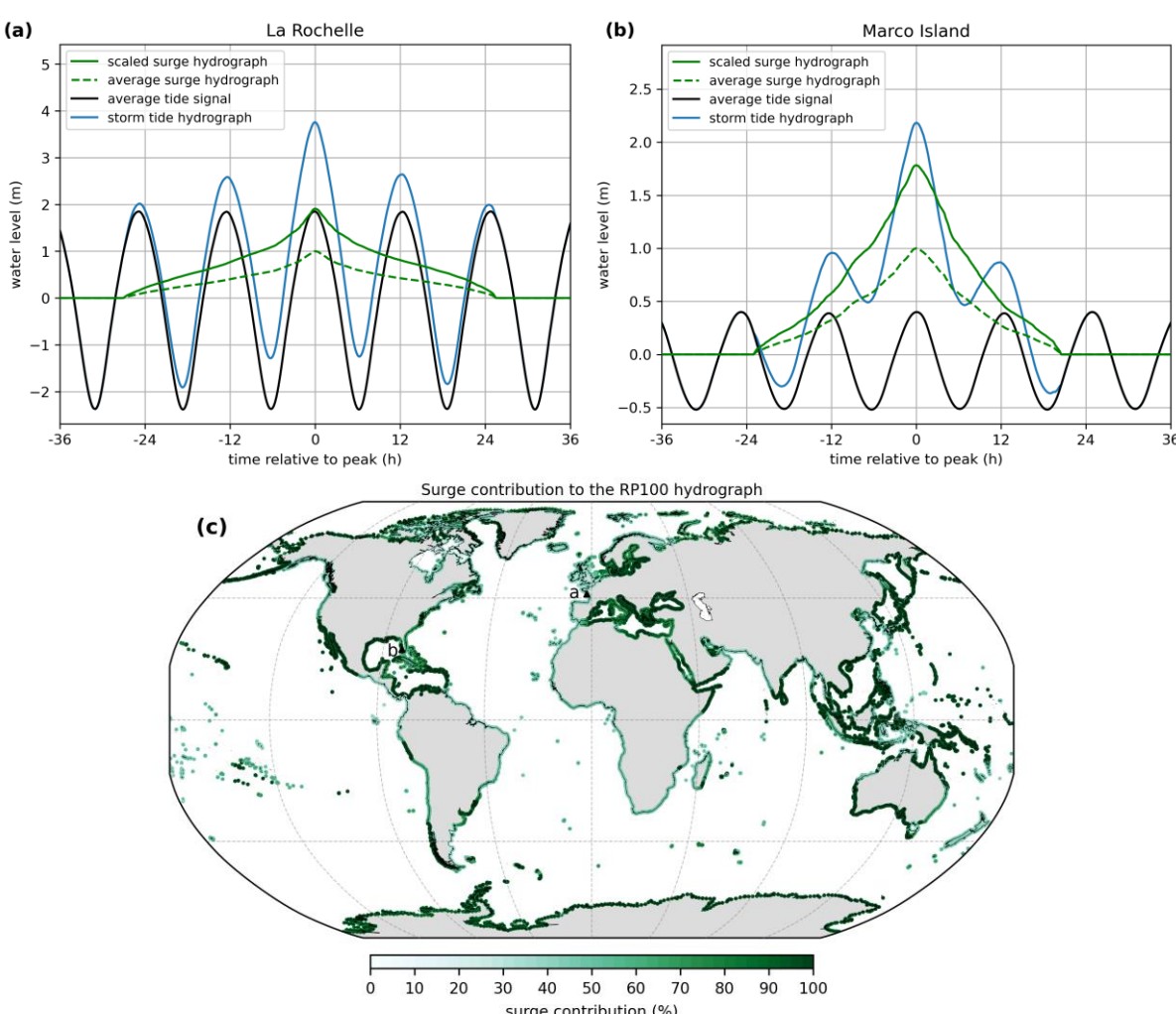

*Figure 6: RP100 storm tide hydrograph (blue line) for a) La Rochelle and b) Marco Island. The average tide signal (black line), average surge hydrograph (green line), and scaled surge hydrograph (dashed green line) are also shown. Panel c) shows the relative contribution of the surge to the RP100 storm tide hydrograph maximum as a percentage. The locations of La Rochelle and Marco Island are indicated by the letters a and b, respectively.*

## 4.4 Assumptions underlying the hydrograph

HGRAPHER is based on certain assumptions to create the storm tide hydrographs. Here, we aim to better understand how these assumptions influence the results. First, we assume that the POT99 threshold results in the selection of a set of surge events from the residual time series that represents the typical evolution of a surge event at any location. However, using a higher or lower POT percentile to select surge events will give different results, depending on the typical characteristics of a location. We illustrate this using La Rochelle and Marco Island as an example. Using a higher (POT99.5) or lower (POT98) POT percentile does not result in a clearly deviating surge hydrograph at La Rochelle (*Fig. 7a*). At Marco Island however (*Fig. 7b*), a clear difference can be observe d between the surge hydrographs. Using the higher POT99.5 percentile (i.e. only using the ~20 most extreme surge events) results in a hydrograph that is more narrow and has a shorter duration. This is most likely caused by the different types of storms that occur at Marco Island. Using a higher POT percentile as threshold will result in an event set with a relatively larger share of TCs, compared to ETCs. This indicates that surge events caused by TCs are typically shorter compared to ETC-related surge events at Marco Island. Wahl et al. (2011) also showed that the peak of the surge hydrograph can show a dependency to the intensity of the underlying surge events. At the global scale, it can be observed that the surge hydrograph duration

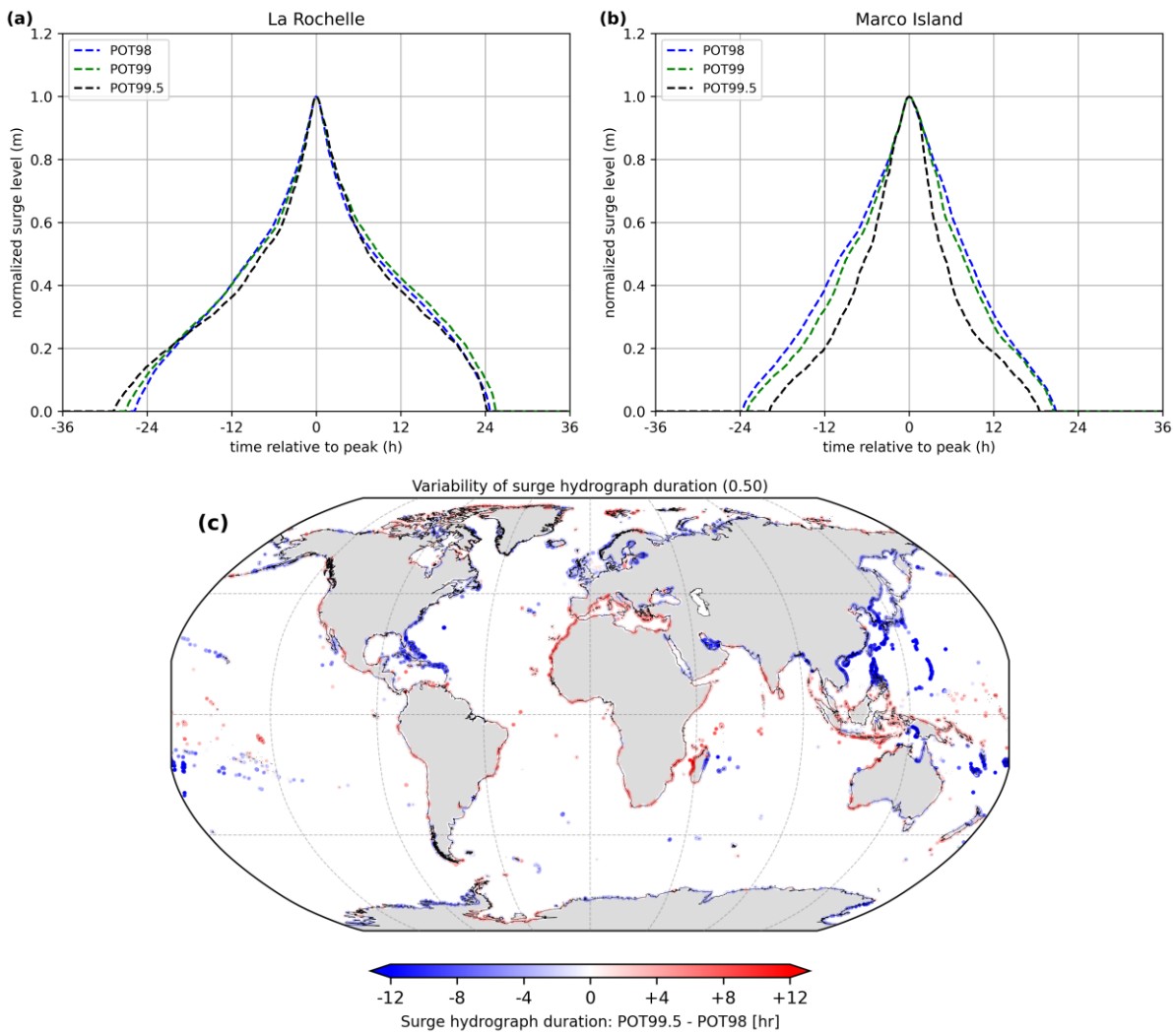

*Figure 7: POT99 surge hydrograph (dashed green line) for a) La Rochelle and b) Marco Island. The blue and black dotted lines show the average surge hydrograph based on the surge events that exceed the POT98 and POT99.5 percentile. Panel c) displays the difference in surge hydrograph duration in hours at a normalized surge level of 0.5, computed as POT99.5 minus POT98.*

(at the unitless 0.5 level) is typically shorter in the Caribbean and northwest Pacific Ocean when only
using the more extreme surge events (i.e. POT99.5 relative to POT98) for generating a surge
hydrograph (*Fig. 7c*). Outside TC prone areas the variability in surge hydrograph duration, either
positive or negative, is less pronounced. Overall, to select the best POT percentile to generate the
surge hydrograph, knowledge about the local conditions is required. For example, if surge events
happen very infrequently (i.e. less than once per year) a percentile higher than POT99 should be used.
Correspondingly, in areas where TCs occur such a higher POT percentile should be chosen if the
research focusses on TCs. For this, knowledge about the number of historical TC storm surge events is
required. Conversely, if the goal is to create a RP1 storm tide hydrograph a lower POT percentile is
warranted compared to when one is interested in the RP100 storm tide hydrograph.
Second, when combining the surge hydrograph with the average tide signal we assume that the
maxima of the two coincide in time. Including a time offset will lead to a storm tide hydrograph of
which the water level is elevated over a longer period of time, potentially increasing the severity of a
flood event. To test this assumption we compute the time offset at La Rochelle and Marco Island (*Fig.*
*8a & 8b*), which is defined as the timing of the maximum storm tide relative to astronomical high tide
(*Fig. 1*). What can be observed at both output locations is that the distribution is centred around zero.
However, at Marco Island the distribution is more spread out, indicated by a standard deviation of
0.68 compared to 0.13 for La Rochelle. At the global scale, large mean absolute time offsets are
observed in areas with either a very small tidal range, such as the Baltic Sea and the Mediterranean
Sea, or a diurnal tide regime in combination with large TC induced storm surges, such as the Gulf of
Mexico (*Fig. 8c*). We show the absolute time offset instead of the actual values because this way all
areas where large time offsets occur are revealed, including areas with both positive and negative
time offsets. The globally averaged absolute mean offset is 33 minutes, and the median is 9 minutes.
To conclude, the assumption that the surge and tide maxima coincide is appropriate at most output
locations. However, at certain locations it should be considered to include a time offset when creating
a storm tide hydrograph.

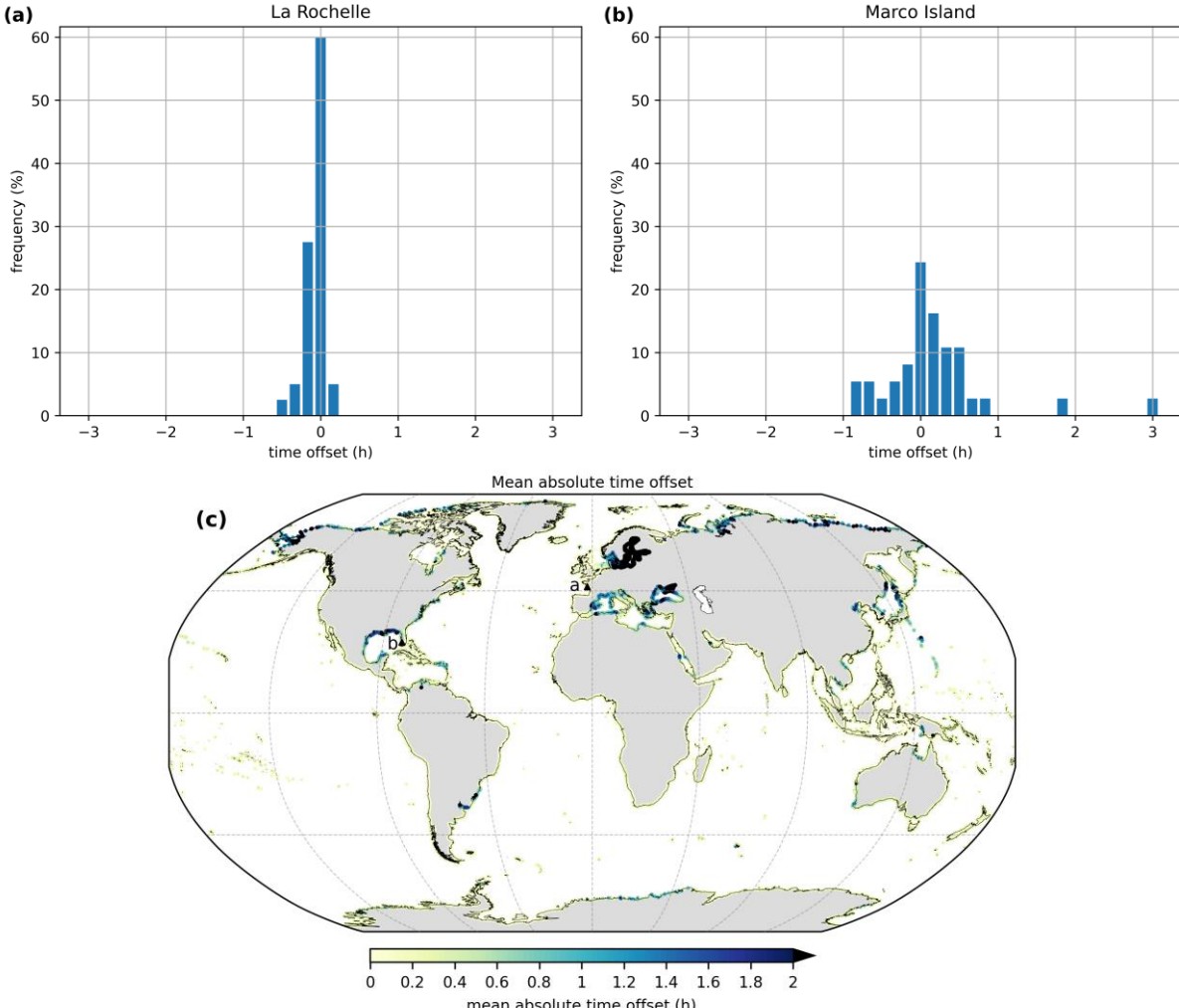

*Figure 8: time offset distribution of the POT99 storm tide maxima relative to astronomical high tide at a) La Rochelle and b) Marco Island. Each blue bar represents a 10-minute period. Panel c) shows the mean absolute time offset in hours. The locations of La Rochelle and Marco Island are indicated by the letters a and b, respectively.*


# 5 Discussion and conclusion

This study improves the understanding of the duration and shape of extreme sea level events along the global coastline. It provides a novel global dataset of storm tide hydrographs which is an important first step in moving away from the planar approach towards dynamic inundation modelling. The open-source HGRAPHER model can generate hydrographs and allows users to create storm tide hydrographs for a RP of interest. Here, we used time series of surge, tide and storm tide levels from the CoDEC dataset (Muis et al., 2020) as input, and generated storm tide hydrographs with a 1-in-100 year return period based on COAST-RP (Dullaart et al., 2021b). Users have multiple options including, 1) use the average tide signal or spring tide signal; 2) define a POT percentile to select surge events for generating the surge hydrograph; 3) include a time offset for combining the surge hydrograph with the tide; 4) define for which RP a storm tide hydrograph should be generated; and 5) use other time series or return periods as input data for HGRAPHER.

Several aspects of our methodology could be further improved. First, we use 38-year of surge level time series that are obtained by subtracting tidal level time series from the storm tide level. As a result, the surge time series do not only contain the meteorological contribution to the sea level, but also contain tide-surge interaction effects (Horsburgh and Wilson, 2007). This could be addressed by using a 'surge-only' simulation, which would not be affected by interaction effects. Another aspect that could be improved is that our analysis is based on a 38 years timeseries. This provides a limited number of events, specifically for regions that do not regularly experience extremes such as the equatorial regions. Potentially we could extend our analysis by using a large set of synthetic events, such as those presented for TCs in (Dullaart et al., 2021a). For extra-tropical regions, seasonal forecasts could be used to create a large ensemble of events (Haarsma et al., 2016). The advantage of a large set of synthetic events is that it would allow to assess if hydrographs are different for different RPs. This is currently not possible because of the small sample size.

Second, we do not account for different types of storms. TCs and ETCs have distinct meteorological characteristics resulting in a different evolution of the water level over time. For example, TCs can have stronger wind speeds and lower air pressure than ETCs, resulting in a higher storm surge (Keller and DeVecchio, 2016). ETCs on the other hand generally affect a larger coastal area because they are often larger in size than TCs (Irish et al., 2008). The typical radius of a TC is between 100 and 500 km while for an ETC it is in the range of 100-2000 km. In addition, once TCs move inland the wind direction can become offshore directly at the coast, resulting in a storm surge sign that quickly changes from positive to negative. An example of this is during TC Irma, which made landfall in Florida in 2017 (Cheng and Wang, 2019). A potential direction for future research would be to separate storm surges by type of storm that caused them and develop a surge hydrograph individually for TCs and ETCs. This would require much longer surge time-series (representing thousands of years instead of decades) that could be created using, for example, large climate model ensembles (Haarsma et al., 2016) or synthetic tracks of TCs (Bloemendaal et al., 2020).

Third, the average tide signal is computed by taking the average over thousands of tidal cycles with a duration of 1 lunar day, lasting 24 hours and 50 minutes. For the majority of the output locations HGRAPHER correctly extracts the average (spring) tide signal. However, in areas with a mixed tidal regime the daily uneven magnitude of the two high tides are averaged out. This is because over time it alters whether the first or second high tide is the highest tide during that lunar day. Including the mixed tidal regime characteristics at these locations, such as Montevideo (*App. Fig. A2*), would result in a more realistic storm tide hydrograph. However, for this multiple storm tide hydrographs have to

be generated that have different shapes but reach the same maximum water level. This would make
the storm tide hydrographs dataset less easily applicable in large scale flood hazard assessments.
A final limitation is that our analysis does not include waves. Wave setup can increase storm tide levels
at the coast. Therefore, it is often an important component of extreme sea levels, and including a
dynamic wave setup component in HGRAPHER is a potential direction for future research. To
accomplish this, we could make use of a parametric approach that has been used in previous global
scale studies to obtain estimates of wave setup (Vousdoukas et al., 2018; Kirezci et al., 2020).
HGRAPHER and the global dataset of storm tide hydrographs improve our understanding of the
duration and shape of storm tide levels. They provide a basis to move towards more dynamic
inundation modelling across different spatial scales, and as a next step, the hydrographs could be
applied as boundary conditions in inundation modelling. This way, the time component is taken into
account when modelling inundation which will substantially improve the accuracy of coastal flood
hazard assessments across different spatial scales.
# 6 Appendices

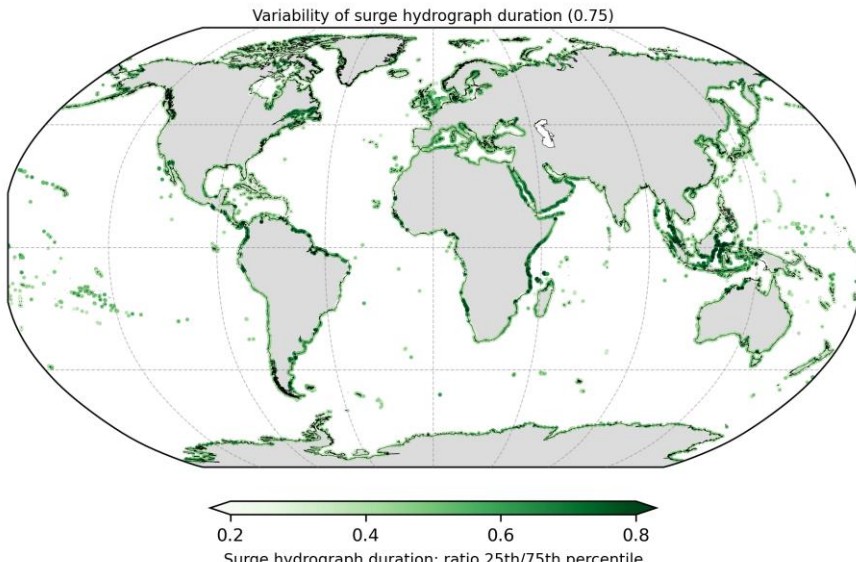

*Figure A1: Ratio of the surge hydrograph duration of the 25th and 75th percentile at the normalized surge height 0.75. The ratio is computed by dividing the 25th percentile value by the 75th percentile value.*


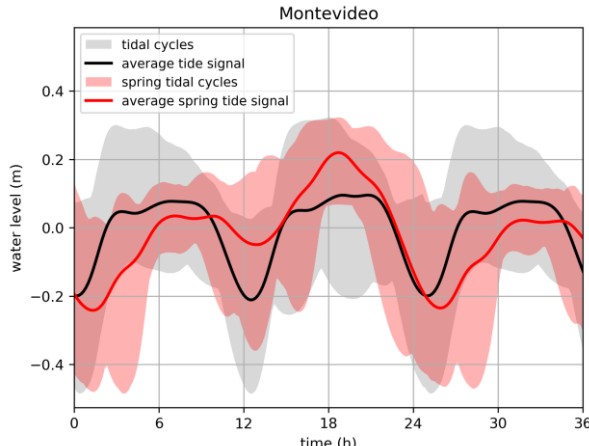

*Figure A2: Average tide signal (black line) for Montevideo. The grey shaded area shows the range of all tidal cycles. The average spring tide signal is shown in red and the red shaded area indicate all tidal cycles that are used to compute the average spring tide signal.*


# 7 Code availability

The HGRAPHER method developed in this study consists of several scripts that are available from GitHub at github.com/jobdullaart/HGRAPHER

# 8 Data availability

Sea level time series used in this study are available from the Copernicus Climate Data Store (CDS) at doi.org/10.24381/cds.a6d42d60. In addition, the hydrographs generated in this study are available from the 4TU data repository at doi.org/10.4121/21270948.

# 9 Author contribution

JD developed the HGRAPHER method and wrote the paper. SM, HM, DE, PW, and JA participated in technical discussions and co-wrote the paper.

# 10 Competing interests

The authors declare that they have no conflict of interest.

# 11 Acknowledgements

We would like to thank Nathalie van Veen for her active involvement in the interpretation of the model outcomes and her critical look at the methodology. J.D. and J.A. received funding from the COASTRISK project financed by the SCOR Corporate Foundation for Science (R/003316.01). J.A. is also funded by the ERC Advanced Grant COASTMOVE #884442. S.M. received funding from the research program MOSAIC with project number ASDI.2018.036, which is financed by the Netherlands Organization for Scientific Research (NWO). D.E. and P.W. received funding from the NWO in the form of a VIDI grant (grant no. 016.161.324). This work was sponsored by NWO Exact and Natural Sciences for the use of supercomputer facilities (grant no. 2020.007).

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
