# Peer review of "Enabling dynamic modelling of global coastal flooding by"

_EGUsphere, 2022_

## Referee Comment (RC1)

**Review - Enabling dynamic modelling of global coastal flooding by defining storm tide hydrographs**

**Summary**

The article by Dullaart et al. presents a method for the generation of storm tide hydrographs on a global scale using a new tool called HGRAPHER. Building on previous work by Chbab (2015), HGRAPHER generates storm tides for specific return periods specified by the user.

The paper is generally well-written and the methods described are resonably justified. While improvements to the work can be made, these are identified and presented by the authors. The authors state the work represents a first step to bringing storm tide hydrographs to global analyses of coastal flooding using hydrodynamic models, and I agree. I recommend acceptance of this article after some revision.

**General comments**

- While I had some initial comments regarding the handling of the hydrograph temporal evolution, much of these were discussed in Section 6. While it is suggested in the manuscript that the storm tide duration can influence flooding, I found no references to this fact. Perhaps the authors could include either Santamaria-Aguilar et al. (2017: https://doi.org/10.1002/2016JC012579) or Quinn et al. (2014: https://doi.org/10.1002/2014JC010197) in their explanation of why storm tide duration should be considered?

**Specific comments**

(Line numbers are specified for each comment)

**Abstract**

11. This first sentence makes me think that coastal flooding can occur under high tides alone, which is not the case. I think the use of "or" implies that

storm surges are not required to drive coastal flooding.

12. tropical and extratropical ... cyclones?

**1. Introduction**

27. "as a result of increasing exposure" - increased exposure is the result of physical and socioeconomic changes, not the other way around.

**2. Available methods to generate hydrographs**

156. While I understand that the method by MacPherson et al. (2019) is not applicable on a global scale, it is still applicable at larger scales, including the entire Baltic Sea and other regions of low tides.

**3.2.2 Average and spring tide signal**

220. I would like a bit more clarification on this point. You extract all tidal cyles of 24 hours and 50 minutes (presumably because this is the phase of the M2 tidal component?) but I am not sure what this really entails. Do you split the tidal series up into segments that are each 24 hours and 50 minutes long, and take the mean of all these segments? Then in figure 3b there are tidal signals that are 72 hours in length. Are these related? I think a clearer description of this process is needed.

**4.2. Average (spring) tide signal**

288. - 292. Regarding the choice of maximum average or spring tide, I am not sure why a random tide is not considered. The example given is that in northwestern Australia, the spring tide is much larger than the average maximum tide, and therefore an extreme storm tide is more likely to occur during a spring tide. However, this ignores the fact that spring tides occur less often than tides of height equal to the average maximum, and that the region is prone to tropical cyclones which can cause storm surges significantly larger than events produced by extratropical events. What is unclear to me, is why a simple statistical analysis of tides was not performed, providing a distribution of tidal water levels at the time of the storm tide maximum? HGRAPHER could then produce a tidal signal of a given height, rather than rely on either the average maximum of spring tide. I can only think that the authors wanted to produce events with similar tidal regimes spatially and across different return water levels. If this is the case, it should be stated in the methods.

**4.4 Assumptions underlying the hydrograph**

311. - 330. This is an important paragraph which answers much of my qestions regarding the performance of the method in simulating the storm tide temporal evolution. The authors state the choice of threshold could be used to better model events of specific heights (i.e. TC events can be better modelled with higher thresholds, lower events with lower thresholds). I would be interested in the performance of the model if a double threshold approach was considered, where a lower threshold is used to rule out events below a desired level and an upper threshold is introduced to rule out events above a certain level. For example, if I was interested in a RP100 water level at some specific site, perhaps I could set a lower threshold equal to RP100-0.25m and and an upper threshold equal to RP100+0.25m. This would ensure HGRAPHER only considers events equal in magnitude to my desired water level.

---

## Author Comment (AC1)

**Review response (#1): Enabling dynamic modelling of global coastal flooding by defining storm tide hydrographs**

Job C. M. Dullaart, Sanne Muis, Hans de Moel, Philip J. Ward, Dirk Eilander, and Jeroen C. J. H. Aerts

**Reviewer #1 (Summary):**

The article by Dullaart et al. presents a method for the generation of storm tide hydrographs on a global scale using a new tool called HGRAPHER. Building on previous work by Chbab (2015), HGRAPHER generates storm tides for specific return periods specified by the user. The paper is generally well-written and the methods described are reasonably justified. While improvements to the work can be made, these are identified and presented by the authors. The authors state the work represents a first step to bringing storm tide hydrographs to global analyses of coastal flooding using hydrodynamic models, and I agree. I recommend acceptance of this article after some revision.

**Authors' response**

*We would like to thank the reviewer for the time taken to review our manuscript. We are pleased to read that the reviewer considers the manuscript to be well-written and the presented work a valuable first step to bringing storm tide hydrographs to global analyses of coastal flooding using hydrodynamic models. Following the reviewer's suggestions, we have revised our manuscript. We feel that these revisions have greatly improved our manuscript. In the following sections we respond to each of the reviewer's remarks or questions. Our response is in italic.*

**Reviewer #1 (General comments):**

While I had some initial comments regarding the handling of the hydrograph temporal evolution, much of these were discussed in Section 6. While it is suggested in the manuscript that the storm tide duration can influence flooding, I found no references to this fact. Perhaps the authors could include either Santamaria-Aguilar et al. (2017:https://doi.org/10.1002/2016JC012579) or Quinn et al. (2014:https://doi.org/10.1002/2014JC010197) in their explanation of why storm tide duration should be considered?

*We thank the reviewer for suggesting these relevant references. We added the suggested references to the introduction section and included a more detailed explanation of how storm tide duration can influence coastal flooding. The introduction now reads as follows:*

> *L62: "Hydrograph characteristics that determine the flood severity are, among others, the maximum storm tide level, base duration, and overall shape. For example, when the water level is elevated for a longer period of time, particularly when it is close to the time of high water when defence exceedance is most likely, the water will propagate further inland (Santamaria-Aguilar et al., 2017; Quinn et al., 2014)."*

**Reviewer #1 (Specific comments):**

**Abstract**

L11: This first sentence makes me think that coastal flooding can occur under high tides alone, which is not the case. I think the use of "or" implies that storm surges are not required to drive coastal flooding.

*Thank you. Coastal flooding caused by high tides alone has been the topic of some recent studies (e.g. Hino et al., 2019; Thompson et al., 2021). However, these studies focus on the future when SLR will substantially increase the number of locations that experience recurrent high-tide flooding. However, in several parts of the world, so-called nuisance floods do already occur due to high tides alone, such as during king tides – for example, this is a regular phenomenon in Jakarta but also other regions. However, because coastal flooding generally occurs under high tide and storm conditions, we decided to follow the reviewer's suggestion. The text now reads as follows:*

> *"Coastal flooding is driven by the combination of (high) tide and storm surge, the latter being caused by strong winds and low pressure in tropical and extratropical cyclones."*

L12: tropical and extratropical … cyclones?

*We thank the reviewer for their careful reading and have adjusted the text accordingly.*

**1. Introduction**
L27: "as a result of increasing exposure" - increased exposure is the result of physical and socioeconomic changes, not the other way around.

*We agree with the reviewer. This line now reads as follows:*

> *L27: "In addition, the number of people living in coastal areas below 10 m elevation worldwide is projected to increase from over 600 million people today to more than 1 billion people by 2050 under all Shared Socioeconomic Pathways scenarios (Merkens et al., 2016), which means that the exposure will increase."*

**2. Available methods to generate hydrographs**
L156: While I understand that the method by MacPherson et al. (2019) is not applicable on a global scale, it is still applicable at larger scales, including the entire Baltic Sea and other regions of low tides.

*The reviewer raises a valid point here. We included the example provided by the reviewer in the text which now reads as follows:*

> *"In addition, MacPherson et al., (2019) developed a method that is applicable in areas with a small tidal range, making it well suited for the German Baltic Sea coast and larger scales such as the entire Baltic Sea, but inapplicable at the global scale."*

**3.2.2 Average and spring tide signal**
L220: I would like a bit more clarification on this point. You extract all tidal cycles of 24 hours and 50 minutes (presumably because this is the phase of the M2 tidal component?) but I am not sure what this really entails. Do you split the tidal series up into segments that are each 24 hours and 50 minutes long, and take the mean of all these segments? Then in figure 3b there are tidal signals that are 72 hours in length. Are these related? I think a clearer description of this process is needed.

*As correctly stated by the reviewer, we split the tidal series up into segments that are each 24 hours and 50 minutes long. This because this is indeed the phase of the M2 tidal component, equivalent to the duration of a lunar day which is the time of the rotation of the earth with respect to the moon. Indeed, subsequently we take the mean of all these segments to obtain what we refer to as 'the average tide signal'. Concerning figure 3b, where a 72 hours average tide signal is shown, this duration is chosen because the mean of the tidal segments has a length of 24 hours and 50 minutes which is too short to combine the tide with the 72-hours surge hydrograph. Therefore, we duplicate the average tide segment to create a longer tidal time series of 72 hours. We have clarified this in the methods section; the respective paragraph now reads as follows:*

> *"Next, we combine the surge hydrograph with the average tide signal (Fig. 3b). To create a curve representing the average tide signal we take three steps. First, we split the tidal series from the period 1980-2017 up into segments that are each 24 hours and 50 minutes long. The start and end times of the tidal segments are selected from the tide time series by searching for a minimum around 24h and 50 minutes after the previous low tide. The segment length is based on the phase of the M2 tidal component which is equal to a lunar day (24 hours and 50 minutes). At most locations around the world M2 is the main tidal component. Second, we compute the mean over all tidal segments to obtain the average tide segment. Third, we duplicate the average tide segment to create a longer tidal time series of 72 hours, which we refer to as the average tide signal."*

**4.2. Average (spring) tide signal**

L288 – L292: Regarding the choice of maximum average or spring tide, I am not sure why a random tide is not considered. The example given is that in northwestern Australia, the spring tide is much larger than the average maximum tide, and therefore an extreme storm tide is more likely to occur during a spring tide. However, this ignores the fact that spring tides occur less often than tides of height equal to the average maximum, and that the region is prone to tropical cyclones which can cause storm surges significantly larger than events produced by extratropical events. What is unclear to me, is why a simple statistical analysis of tides was not performed, providing a distribution of tidal water levels at the time of the storm tide maximum? HGRAPHER could then produce a tidal signal of a given height, rather than rely on either the average maximum of spring tide. I can only think that the authors wanted to produce events with similar tidal regimes spatially and across different return water levels. If this is the case, it should be stated in the methods.

*While developing HGRAPHER, we considered the reviewer's suggestion of using a random tide. However, we decided to use the average tide signal because the goal of this study is to enable the dynamic approach and move away from the bathtub approach for large-scale inundation modelling. With a bathtub approach, a flood map is created that corresponds to a single water level (e.g. the 1-in-100 year return period). By creating hydrographs, the time component, i.e. the duration of the peak water levels, can be taken into account as well. Randomly selecting tidal levels would result in a large set of possible storm tide hydrographs that all have the same maximum water level. However, to be able to apply this method for large-scale flood modelling, we think an approach based on one flood map per return period is most appropriate.*

**4.4 Assumptions underlying the hydrograph**

L311 - L330: This is an important paragraph which answers much of my questions regarding the performance of the method in simulating the storm tide temporal evolution. The authors state the choice of threshold could be used to better model events of specific heights (i.e. TC events can be better modelled with higher thresholds, lower events with lower thresholds). I would be interested in the performance of the model if a double threshold approach was considered, where a lower threshold is used to rule out events below a desired level and an upper threshold is introduced to rule out events above a certain level. For example, if I was interested in a RP100 water level at some specific site, perhaps I could set a lower threshold equal to RP100-0.25m and an upper threshold equal to RP100+0.25m. This would ensure HGRAPHER only considers events equal in magnitude to my desired water level.

*Using different threshold approaches and evaluating how they influence the model performance is an interesting suggestion. However, we have only 38 years of surge time-series. Reducing the number of surge events on which the surge hydrograph is based may increase the uncertainty. Selecting surge events by using different thresholds as suggested by the reviewer would result in a different number of surge events per location, which we believe would make the methodology spatially less consistent. Yet, as already mentioned in the text (section 4.1), we do think that TCs have a distinct hydrograph shape (mainly a shorter base duration) and a more in-depth analysis of appropriate thresholds for different environmental settings is an interesting avenue for future research. This would require much longer surge time-series (representing thousands of years instead of decades) that could be created using, for example, large climate model ensembles* (Haarsma et al., 2016) *or synthetic tropical cyclones* (Bloemendaal et al., 2020). *We included the latter sentence as a recommendation in the discussion section of the revised manuscript.*

---

## Author Comment (AC2)

**Review response (#2): Enabling dynamic modelling of global coastal flooding by defining storm tide hydrographs**

Job C. M. Dullaart, Sanne Muis, Hans de Moel, Philip J. Ward, Dirk Eilander, and Jeroen C. J. H. Aerts

**Reviewer #2 (Summary):**

The manuscript present a methodology to generate storm tide hydrographs at global scale by combining tidal cycles (average and spring) with an average storm surge hydrograph. The average storm surge hydrograph is developed by selecting first extreme storm surges, normalizing the time series of these events, calculating their average value at each time step and the duration before and after the storm peak. On the other hand, two average tidal cycles are calculated, namely the average and spring tidal cycles. The effect of the tide-surge interaction can also be included when combining the storm surge and tidal hydrographs by including the mean time offset between the two peaks.

The manuscript is well written and structured; the topic is relevant and the results and findings are interesting and relevant. However, I think that there are a couple of points that should be addressed and I also have some minor comments.

**Authors' response**

*We would like to thank the reviewer for the thorough review and useful comments, which we believe have helped us greatly to improve our manuscript. We are pleased to read that the reviewer considers the manuscript to be well-written and structured, and the presented work to be interesting and relevant. Following the reviewer's suggestions, we have revised our manuscript. In the following sections we respond to each of the reviewer's remarks or questions. Our response is in italic.*

**Reviewer #2 (Main comments):**

My main concern with this methodology is that it only focus on generating average hydrographs and thus the variability at other time steps rather than the peak is neglected/lost. Based on the examples that the authors show, it can be seen that historic storm surge hydrographs and tidal cycles show a large variability at times before and after the peaks. The variability of the water levels at times before and after the peak can largely affect the resulting flooding (see e.g. Quinn et al., 2014; Santamaria-Aguilar et al., 2017). The authors briefly discuss this issue regarding TCs, ETCs and mixed semidiurnal tidal regimes in the discussion, but I miss a more detailed discussion or quantification of the variability of the hydrographs related to the mean. For example, how much can vary a hydrograph in a region with a mixed semidiurnal regime (large tidal variability) and affected by TCs and ETCs? Are there regions where the variability of the hydrograph can largely exceed the one of the peak?

*As correctly stated by the reviewer, the focus of the methodology is indeed on generating average hydrographs. We focus on average hydrographs because the goal of our study is to enable large-scale flood modelling using dynamic models. The most common approach for flood risk assessments is to compute flood maps corresponding to different return periods (1 in 10-year, 1 in 100-year, etc.). Having only one hydrograph per return period makes the storm tide hydrographs dataset easy to implement in large scale flood hazard assessments, which are computationally expensive. Another reason for not focusing on the variability of the surge levels is that we only have 38 years of data (resulting in a set of ±38 events). Studying the variability that is caused by specific storm characteristics such as storm length, intensity, and wind direction, would require much longer time. To get some idea of the*

*variability of the hydrographs, we computed the 25th and 75th percentiles surge hydrographs (Fig. 4a & 4b), and we show all tidal segments that are used to calculate the average and spring tide signal (Fig. 5a & 5b). The variability of the water level at times before and after the peak can indeed largely affect the resulting flooding. Therefore, we have added the two references mentioned by the reviewer in the introduction that now reads as follows:*

> *L62: "Hydrograph characteristics that determine the flood severity are, among others, the maximum storm tide level, base duration, and overall shape. For example, when the water level is elevated for a longer period of time, particularly close to the time of high water when defence exceedance is most likely, the water will propagate further inland (Santamaria-Aguilar et al., 2017; Quinn et al., 2014)."*

The authors state that one of the main objectives of this study is to enable dynamic flood modeling at global or large scales by providing storm tide hydrographs. I differ on this as I think that the main limitation for dynamic flood modeling at global scales is still the computational effort required, even by simplified hydrodynamic models like LISFLOOD and SFINCS. However, this methodology contributes to dynamic flood modeling at any other scale as hydrographs are commonly neglected in EVA. Therefore, I think this work would benefit by changing the focus to the limitations of EVA models that neglect the hydrograph information, regardless of the scale of the flood assessment.

*We believe we are on the same line as the reviewer when it comes to the motivation behind this study: we would like to move away from the bathtub approach that is based on an EVA analysis of static water levels and subsequently computing flood maps for various return water levels (e.g. RP100). In this study our aim is to develop a relatively simple method to generate storm tide hydrographs such that it can be applied at large scales. It is true that currently simplified hydrodynamic models are not yet applied at the global scale due to the high computational costs, but there are some examples at the continental scale* (Vousdoukas et al., 2016; Paprotny et al., 2018)*. Moreover, globally applicable approaches are being developed  (e.g. Eilander et al., 2022), which allow for the rapid set-up of inundation models anywhere on the globe, forced by global datasets (or more local data where available) - also for this approach, data on hydrographs are required to move past the bathtub approach. When modelling inundation at the regional to local scale, it would make sense to further advance our methodology such that it is able to incorporate more local characteristics of extreme sea levels, such as was done by Macpherson et al. (2019) for the German Baltic Sea coast. Therefore, we stick to the research gap as described in the original manuscript. However, we adjusted the scale from 'global-scale' to 'continental- to global-scale' throughout the revised manuscript.*

**Reviewer #2 (Minor comments):**

L11 - Coastal flooding can also arise from other drivers such as waves, precipitation, and river discharge in estuarine regions or a combination of all these drivers.

*We changed the first sentence of the introduction as follows:*

> *L11: "Coastal flooding is driven by the combination of (high) tide and storm surge, the latter being caused by strong winds and low pressure in tropical and extratropical cyclones."*

L12- Missing "cyclones" at the end of the sentence.

*Amended*

L16 & L24- At global scale, the main constrain regarding the flood model used (static vs dynamic) is the computational effort. Even simplified hydrodynamic flood models such as LISFLOOD or SFINCS still require huge computational resources compared to bathtub models. In my opinion, the main focus should be placed on losing the hydrograph information when doing the EVA, as most EVA models focus only on storm surges peaks. Therefore, information about the hydrograph or temporal evolution of the event is lost although it is required as boundary conditions for dynamic flood models, even at local scales.

*See previous author comment under 'main comments'*

L27- Exposure increases because of the increase in population, not the opposite way.

*We agree with the reviewer. This line now reads as follows:*

> *L27: "In addition, the number of people living in coastal areas below 10 m elevation worldwide is projected to increase from over 600 million people today to more than 1 billion people by 2050 under all Shared Socioeconomic Pathways scenarios (Merkens et al., 2016), which means that the exposure will increase."*

L30- Global coastal flood risk assessments can help identifying…

*Amended*

L32-Coastal flooding is generally driven by "storm surges" generated from strong winds…

*Amended*

L47- Authors can cite here Ramirez et al., 2016; Vousdoukas et al., 2016

*These studies are now cited in the revised manuscript.*

Some lines of the introduction do not connect very well (are not smooth) e.g. L30 to L35. I'll suggest revising it trying to better connect the points between sentences.

*To improve the flow of the text we made some changes to the first paragraph of the introduction:*

> *L26: "Over the course of the 21$^{st}$ century, coastal populations will be increasingly at risk of flooding due to sea level rise (SLR) (Oppenheimer et al., 2019). In addition, the number of people living in coastal areas below 10 m elevation worldwide is projected to increase from over 600 million people today to more than 1 billion people by 2050 under all Shared Socioeconomic Pathways scenarios (Merkens et al., 2016). Global coastal flood risk*

*assessments can help identifying areas that are potentially exposed to flooding under both current and future climate conditions (Ward et al., 2015). To setup these flood risk assessments, it is important to understand the dynamics of storm surges generated from strong winds from both low pressure in tropical- (TCs) and extratropical cyclones (ETCs) and how these generate coastal flooding (Resio and Westerink, 2008). Flood models can be used to model these coastal inundation dynamics resulting from extreme storm tides, where the storm tide is defined as the combination of storm surge and the tide (Colle et al., 2010)."*

L88- I think authors should cite all different methods available in the literature (or most of them) even if they only describe in more detail a subset of them.

*In the revised manuscript, we now include a much larger number of relevant studies, including (Sebastian et al., 2014; Chbab, 2015; Environment Agency, 2018; MacPherson et al., 2019; Vousdoukas et al., 2016a; Xu and Huang, 2014; Salisbury and Hagen, 2007).*

L100-Why that threshold and duration are selected? Was a sensitivity analysis performed in Chbab (2015)? I think this point has to be mentioned and the sensitivity of the approach to these parameters (i.e. threshold and duration of the event) discuss. In addition, several normalized hydrographs are generated from this approach, which one is used for estimating any desired RP hydrograph? The mean normalized hydrograph? How the variability of these extreme hydrographs can affect the resulting design hydrograph? See e.g. Santamaria-Aguilar et al., (2017) and Quinn et al., (2014)

*We thank the reviewer for pointing out that the description of the method from Chbab (2015) was lacking some details. We have included any missing relevant information (i.e. the selected threshold and duration). Unfortunately, in doing so we found a typo in the text. The threshold for a surge event to be selected is not 0.5 m but 1.5 m. The reason for selecting 1.5 m as threshold is that it results in a set of surge events that is large enough to calculate a representative average, but not too large because then too many small surge events would be selected that have a different physical behaviour than higher surge levels and influence the resulting standardized shape. The reason for extracting 48 hours of the surge time series (24 hours before until 24 hours after the surge maximum) is that the duration of a typical storm in the area is also 48 hours. The revised manuscript reads as follows:*

> *L102: "To test the sensitivity of the surge hydrograph to the chosen parameters, a sensitivity analysis performed. They conclude that the upper 50% of the normalized surge height (normalized surge height > 0.5) is not affected when either the threshold or time window length is increased or decreased. This is an important finding because it indicates that the surge hydrograph is most robust close to the time of high water when defence exceedance is most likely (Santamaria-Aguilar et al., 2017; Quinn et al., 2014). However, a longer time window (of e.g. 72 or 96 hours) results in a longer base duration. The argument given for using a 48-hour time window is that 48 hours is the typical duration of a storm along the Dutch coastline.*

L121- Although this is true for most of the places, a dependency between skew surges and high tidal levels is observed at some locations, se Santamaria-Aguilar and Vafeidis (2018).

*To make readers aware that at some locations there can be a dependency between skew surges and high tidal levels, we include the following sentence:*

> *L134: "apart from some locations where a dependency between skew surge and high tidal levels is observed (Santamaria-Aguilar and Vafeidis, 2018)"*

L122- How sensitive is this method (i.e. the 15 events selected) to the series length? Is there any limitation of this approach to a minimum length of data?

*Information about the sensitivity of this method to the series length and the number of events selected (15) is not provided in the report of the Environment Agency. We added the following sentence to the manuscript to inform readers about this:*

> *L137: "An argument for selecting this number of events is not given."*

L157. A similar method than MacPherson et al., (2019) was developed by Wahl et al., (2012) for the German North Sea coast (macrotidal)

*The authors are aware that the study by Macpherson et al. (2019) makes use of the methods developed in Wahl et al. (2012). To make readers aware of this as well the text now reads as follows:*

> *L142: "The third method by MacPherson et al. (2019), that further developed the method from Wahl et al. (2011, 2012), starts by identifying storm tide events."*

L161. The method of Vousdoukas et al., (2016) overestimates the WLs of the hydrograph assuming the maximum high tidal level along the entire duration of the event (i.e. neglecting the tidal variations/cycle), and this issue can significantly overestimate the WLs in those places of large tidal variability (e.g. places with macrotidal range). I think this point needs to be mentioned here or when the approach is described.

*We agree with the reviewer that this limitation of the method from Vousdoukas et al. (2016) could be clarified in the manuscript. Therefore, we included the following sentence:*

> *L154: "The assumption that the maximum high tidal level occurs along the entire duration of the event, thereby neglecting tidal variations, can significantly overestimate the water level in places with large tidal variability, such as north-western Australia."*

L167. I'm missing a bit of a discussion about the variability of hydrographs and why this is important.

*To emphasize the importance of the variability of the hydrograph, particularly close to the maximum water level, we added the following sentence to section 2 of the revised manuscript:*

> *"It is especially important that the hydrograph represents the water level correctly close to high water when defence exceedance is most likely, and because the water will propagate*

*further inland if the water level is elevated for a longer period of time (Santamaria-Aguilar et al., 2017; Quinn et al., 2014)."*

L183-Is this comparison/validation based only on the peaks? How well these modeled data represent the hydrographs?

*This validation study focused on the peaks, or maximum surge levels. To make this clear to the reader we have put the word 'maximum' in front of 'surge heights'.*

L207- i.e. dividing each surge level by the peak

*Amended*

Average tide signal and spring tide signal. Although it is briefly discuss later, I think authors should mention the issues of this approach in places with different tidal regimes, e.g. mixed semidiurnal regimes in which the variability of high tidal levels is large and the spring cycle cannot really be defined?

*In areas with different tidal regimes, such as a mixed semidiurnal regime, and where the variability of high tidal levels large our methodology might not be able to correctly represent the average (spring) tide signal. Indeed it is important to make the reader aware of this. Therefore, this is extensively discussed in the fourth paragraph of the discussion.*

L256. Storms

*Amended*

L262. Differences in the time evolution of storms can also contribute to the differences/variability observed at the different time steps of the hydrographs, not only the effects of the tide-surge interaction. This can be particularly true in those places that are affected by TCs and ECs, as the characteristics of the two types of storms differ, but the authors analyze together the storm surges hydrographs that arise from TCs and ECs. (It is discussed later in section 5, but I think it should be at least mentioned here too).

*The results section now also mentions differences in the time evolution of storms as a potential contributor to the variability observed at the different time steps of the hydrographs. The following line is now included in the results section 4.1 of the manuscript:*

> *L284: "Differences in the evolution of storms over time can also contribute to the variability observed at the different time steps of the normalized surge levels, particularly in areas that are affected by TCs and ETCs as the characteristics of these two types of storms differ considerably* (Domingues et al., 2019)*."*

L283. I would like to see how much the finding are affected by this.

*The large variability between tidal cycles makes it more difficult to extract an average (spring) tide signal that represents the tidal characteristics at this location (Montevideo). In the manuscript we argue that the effect on the combined hydrograph is small because this large variability between tidal cycles is something we observe especially in areas with a mixed semidiurnal tide regime and small tidal range. The high tide value (maximum of the average (spring) tide signal) is however still correctly represented by the hydrograph. Therefore we cannot exactly quantify the effect of this finding on the final hydrograph. However, one can look at the spread between tidal cycles in Figure A1 to get some idea.*

L294. What are the differences at other time steps and their potential effects?

*Generally, when the difference between the average and spring tide signal maximum (high tide) exceeds 0.5 m or 1.0 m, the difference between the average and spring tide signal minima (low tide) also exceeds 0.5 m or 1.0 m. It is mostly the magnitude that differs between the tidal cycles, due to the M2 phase, and not the overall shape (at what time the tide goes up or down). As a result, differences are small between tidal cycles around a water level of 0.0 m, and this difference increases towards the time of high or low tide. Because we are especially interested in maximum water levels, when the water can potentially flood the land, we report the difference between maximum tide levels in the manuscript and not at other time steps. For the two locations that are discussed throughout the paper, the evolution of the average and spring tide signal can be seen in Figure 5a and 5b.*

L330- The previous study of Wahl et al., (2011) showed that the peak of the hydrograph can show a dependency to the intensity of the hydrograph. I think this issue should be mentioned here as e.g. a dependency of the shape of the hydrograph shows a dependency to the threshold in the case of Marco Island.

*It is now mentioned in the manuscript in section 4.4. that Wahl et al. (2011) found some dependency of the peak of the hydrograph to the intensity of the underlying surge events. The study from Wahl et al. (2011) is cited.*

L334- The dependency of the storm surge peak and tidal levels not only can change the duration of the event due to the time offset, but also the magnitude of the event. Not accounting for this time offset overestimates the peak water levels as the NTR peak magnitude might not be the same at high tidal levels (i.e. for the same atmospheric forcing, the NTR magnitude is larger at lower tidal levels than at higher tidal levels). This issue should be mentioned here.

*We agree with the reviewer that in theory we could overestimate the peak water levels as the NTR peak magnitude might not be the same at high tidal levels compared to low tide. However, before combining the tide signal and surge hydrograph, we scale the surge hydrograph up to a certain water level such that the combined water level equals a specific return period (e.g. RP100 based on the COAST-RP dataset). This means that we will never overestimate (or underestimate) the peak value. However, what could be wrong is the contribution of the surge and tide levels to the combined water level. One component might generally be larger (or smaller) in reality, and is one of the limitations of our methods. This is now mentioned at the end of the section 4.3:*

*"Last, the surge hydrographs are based on the surge residual including tide-surge interaction effects. These interaction effects tend to be positive at low tide and negative at high tide (Horsburgh and Wilson, 2007). As a result, we might overestimate the contribution of the surge to the combined hydrograph at high tide."*

---

## Author Response (AR2)

**Review response (2nd round): Enabling dynamic modelling of coastal flooding by defining storm tide hydrographs**

Job C. M. Dullaart, Sanne Muis, Hans de Moel, Philip J. Ward, Dirk Eilander, and Jeroen C. J. H. Aerts

**Reviewer #1 (Comments):**
First of all, I would like to thank the authors for taking time addressing my comments and making the corresponding changes in the manuscript. However, I am not fully satisfied with the little change made to address the limitation of this method to account for the variability of hydrographs at each time step (both tidal and non-tidal) and thus the uncertainties related to the generated mean hydrograph. I believe this is an important factor that can affected the estimated flooding as affects not only the volume of water that can propagate inland, but also other factors such as flood velocities, which can be important for assessing damages and warning systems. I agree with the authors that the short period of the data used also limits the analysis of the variability of extreme storm surge hydrographs. However, I think the authors could have provided a first estimation of this variability at global scale (with not much effort as it is shown for two locations), which could also be used to geographically assess in which areas the variability of hydrographs is large and flood assessments would benefit from including it.

I also still believe that the work presented in this manuscript would have been benefited from providing a different perspective on the lack of information about hydrographs, which is need at all scales for dynamical flood modeling, rather than focusing on continental to global scales, for which the main limitation is still the computational time required to run dynamic flood models.

Saying this, I think the work presented in this manuscript provides a first step on the generation of storm tide hydrographs at global scale and I recommend that it can be accepted.

**Authors' response**
*We would like to thank the reviewer for the time taken to review our manuscript for a second time. We are pleased to read that the reviewer recommends the manuscript to be accepted after revisions. Following the reviewer's suggestions, we have revised our manuscript. We feel that these extra revisions have further improved our manuscript.*

1) *Variability of the hydrographs:*
   *To provide a better global view of the hydrograph variability we have added two figures to the manuscript. Figure 7c shows the difference in surge hydrograph duration (computed as POT99.5-POT98) in hours at a normalized surge level of 0.5. This figure confirms our previous conclusion: the width of the hydrograph in areas prone to tropical cyclones is smaller when using a higher POT threshold for selecting surge events. To explain this we added the following sentences to the manuscript: "At the global scale, it can be observed that the surge hydrograph duration (at the unitless 0.5 level) is typically shorter in the Caribbean and northwest Pacific Ocean when only using the more extreme surge events (i.e. POT99.5 relative to POT98) for generating a surge hydrograph (Fig. 7c). Outside TC prone areas the variability in surge hydrograph duration, either positive or negative, is less pronounced." In addition, Figure A1 (part of the Appendices) shows the ratio of the surge hydrograph duration of the 25th and 75th percentile at the normalized surge height 0.75. The ratio is computed by dividing the 25th percentile value by the 75th percentile value. As suggested by the reviewer, this figure could*

*be used to geographically assess in which areas the variability of the hydrograph is large and flood assessments would benefit from including it. This is also emphasized in the manuscript section 4.1 which now reads as follows: "Last, we computed the difference in surge hydrograph duration between the 25th and 75th percentile at a normalized surge height of 0.75 (App. Fig. A1). This can provide some insights in the variability of flood duration, assuming that inundation might starts to occur around the 0.75 normalized surge height."*

2) *Perspective of spatial scales:*
   *Following the reviewer's suggestion we have changed the perspective of the spatial scale at which our hydrograph method could be applied. Most importantly, we have removed the word 'global' from the title. In addition, we have made multiple adjustments throughout the manuscript to highlight that the developed hydrograph method is not necessarily only applicable at larger scales. Instead, coastal flood modelling assessments at smaller scales would also benefit from including the time component. For example, line 85 (introduction section) now reads as follows: "The aim of this study is to address this research gap by developing and applying a globally-applicable method (HGRAPHER) to generate hydrographs. In doing so, we pave the way for coastal flood mapping using dynamic models across different spatial scales."*